# The Role of Healthcare Professionals in the Transition Process: A Qualitative Study of Families of Trans Children and Adolescents

**DOI:** 10.3390/healthcare12100974

**Published:** 2024-05-09

**Authors:** Lucía González-Mendiondo, Aingeru Mayor, Nieves Moyano

**Affiliations:** 1Departamento de Ciencias de la Educación, Universidad de Zaragoza, 50009 Huesca, Spain; luciam@unizar.es; 2Departamento de e Lenguajes y Sistemas Informáticos, Universidad del País Vasco, 48940 San Sebastián, Spain; aingeru.mayor@ehu.eus; 3Departamento de Psicología Evolutiva y de la Educación, Universidad de Jaén, 23071 Jaén, Spain

**Keywords:** trans children, support, healthcare professionals, transition, family

## Abstract

The aim of this study was to explore the role that healthcare professionals, such as pediatricians, psychologists, psychiatrists and sexologists, play in supporting trans children and adolescents in their transition processes. To this end, 22 life stories written by parents of trans children and adolescents who had undergone social transition were collected, and a qualitative analysis was carried out using MAXQDA software. For this purpose, three key periods were considered in the stories: before, during and after the transition. Among other aspects, the stories highlight a major lack of understanding on the part of the professionals who should support trans children and adolescents, and their families, in their transition processes. Parents described the attitude of those professionals who actively listened to their children’s needs and were cooperative as positive, even if they do not have the necessary training. The need for specific training in diversity is one of the main conclusions of this study.

## 1. Introduction

In recent years, there has been a shift in the way gender identity is understood, and it is now considered to be on a spectrum rather than being a binary phenomenon limited to male and female categories [1]. In fact, many highly diverse ways of expressing gender identity are currently becoming visible, including transgender, non-binary, gender expansive, gender fluid, etc. These gender identities are agglutinated under the term “trans” and share the quality of not corresponding to the sex assigned at birth based on the individual’s genital anatomy. In this study, we focus on the experience of those trans children and adolescents who, at some point in their lives, clearly affirmed to be male or female and have transitioned to live according to their gender identity.

In contrast to non-binary people, who do not typically identify as such until adolescence or early adulthood, trans people’s identity awareness may emerge during the early stages of puberty [2] or even during early childhood [3,4]. Consequently, we are witnessing a growing interest in trans childhood, and the number of studies on the issue of trans children and adolescents has grown exponentially [5]. However, its etiology is still unknown, and we find little agreement between different disciplines; although most of the studies agree that trans childhood is the result of an interaction of various biological, psychological and social factors, ultimately, each study supports either neurobiological or psychosocial theories [6].

Currently, two models of care for trans children coexist. The first is based on the concept that, in early childhood, one’s own identity has not yet been internalized, which becomes perceptible and stable at around 8 years of age and is consolidated during adolescence [7]. This perspective recommends watchful waiting during childhood [8], relying on an allegedly high detransition rate upon reaching puberty (80–95%) [9]. However, the validity of these data has already been undermined [10,11], and the first longitudinal studies on young people who have transitioned in childhood show that only 2.5% subsequently retransitioned [12].

In response to the watchful waiting approach, the affirmative model emerged; this model promotes the notion that families should listen to and affirm the gender identity expressed by their child and attend to their needs, exploring and supporting a process of social transition. The model is based on studies focused on the experience of families of children who made the social transition before puberty and reported positive consequences of social transition, family acceptance and support on their mental health [13,14,15]. According to these studies, the support and acceptance of the child’s gender identity since childhood can favor the construction of a secure attachment and the development of greater resilience [16]. Olson et al. [17], in a study of 73 children aged 3–12 years who had already transitioned, concluded that making the social transition had a positive impact on mental health where, after the transition, the children presented with normative levels of depression and only marginally elevated levels of anxiety. However, their study has been criticized by Schumm and Crawford [18] who, in addition to pointing out methodological flaws, stated that the conclusions were not supported by the data presented; moreover, they reanalyzed the raw data of the study and found that although trans children had only slightly higher levels of depression than cis children, their anxiety levels were significantly higher. Nevertheless, regardless of whether trans children who have transitioned are doing as well as their cis peers, it is most important to determine whether or not making the social transition improves the quality of life of trans children. Recent qualitative research [19] highlighted that distress, sadness, frustration and despair are commonly experienced by children before social transition, as well as feelings of joy on receiving support from their parents, who observed sustained improvements in their children’s mental health, well-being, educational achievement and happiness after their social transition. Accordingly, in recent years, more trans children are being supported with respect to social transition [13].

There is no social consensus on how best to care for trans children and adolescents, and the direction of legislation is variable among countries. Nevertheless, knowledge of the issue is increasing, despite some gaps, and clinical guidelines that provide an evidence-based consensus on health care for trans people have been published [20,21]. Regarding social transition, Standards of Care 8, published by WPATH [20], recommends that parents, caregivers and healthcare professionals respond supportively to children who desire their internal sense of gender identity to be acknowledged.

We can define social transition as an external recognition of the child’s lived gender identity, exploring and supporting a process that entails ceasing to live socially according to the sex assigned at birth, instead living according to one’s gender identity; this may involve a change of name, pronouns, manner of presentation, hairstyle or clothing [18,22]. This transition process is mainly carried out by the environment and, above all, represents a change in the perceptions that others have; it entails ceasing to see the girl who was “supposed” to be in favor of progressively seeing the boy who is, or vice versa [23,24]. Among the difficulties encountered by families who decide to support their children’s transition during childhood, qualitative researchers have pointed out the problem of a lack of knowledge [18,25]. It is common for many of these families to consider what their children express to only be a form of confusion in the moment or a sign of a possible future same-sex sexual orientation [26]; on many occasions, these misconceptions are held by the professionals who support the child, especially in the educational environment [24,27]. Other researchers have pointed out that the main resource that these families turn to in order to alleviate their lack of knowledge and information is the LGBTQIA community and organizations of families of trans children, reached via the internet [28,29].

Throughout the transition, families interact with many healthcare professionals who can play a key role in supporting the transition process [30]. However, it is striking that these professionals are not the main source of information available to the families; in fact, on some occasions, the professionals themselves become a barrier, and both families and the children and adolescents themselves are demanding more training for healthcare professionals [30,31,32,33]. As indicated by Mikulak et al. [34] in a qualitative study involving health professionals working with young trans people, barriers that prevent effective health care can be categorized into four domains: structural (scarcity of specialist centers and long waiting times), educational (lack of inclusion of trans health in professionals’ training), cultural (negative attitudes toward trans people resulting in discrimination within primary care settings) and technical barriers (inflexible information technology systems and gendered spaces).

Against this background, some researchers have urged pediatricians and other physicians to become informed and knowledgeable about the needs of trans children and adolescents, beyond their own political or religious ideologies [35], not only to improve the health status of trans people but also to increase the competence and confidence of healthcare professionals when working with trans people [36].

A recent systematic review by Holland et al. [37] showed how trans adults have very disparate experiences with primary care and emphasizes the need for clinicians and policymakers to focus on the voices of the trans community when designing and improving services.

In order to contextualize the present study, it is important to take into account the progressively greater social and media recognition of gender diversity that resulted, in the Spanish legislative sphere, in the approval in February 2023 of the so-called “Trans Law” (Law for the real and effective equality of trans people and for the guarantee of LGBTQIA rights). This law recognizes the right to gender self-determination, but only from the age of 14 years; this has been undoubtedly one of its most controversial aspects.

In the health field, the first multidisciplinary gender identity unit in Spain was created in 1999 and, in the following years, gender units were progressively implemented in the rest of the autonomous communities. “Gender identity units” are the reference hospital units of the public health services responsible for the health care of the trans population. They are made up of multidisciplinary teams composed mainly of professionals in psychiatry, psychology and endocrinology; usually, they also include the respective pediatric specialists in these fields and, in the case of adults, plastic and reconstructive surgeons. The units follow international health guidelines and update their protocols according to advances that are taking place in different areas. Until very recently, psychiatrists were the gateway to the unit and those who supervised the entire process. In recent years, this model has been changing, albeit not without controversy: it is other professionals from the unit who welcome and accompany users in their transition process, and in some gender units, psychological diagnosis is no longer required. Moreover, some services have been transferred to primary care [38]. In addition, a progressive increase in the number of trans persons seeking specialized treatment has been observed in gender units, along with a notable decrease in their average age [39].

### The Current Study

This study is based on the theoretical framework offered by substantive sexology; this was developed in Spain from the theoretical model of Amezúa [40], which conceives of diversity as a human value to be cultivated and not as a pathology to be cured. This approach assumes that the construction of one’s own identity is not biological *or* cultural, but rather biological *and* cultural. That is, it is a biographical process [41]. Moreover, the current study relies on a transemancipatory theoretical framework [18] and takes into account the role of the pathologization of gender diversity in the maintenance of cisnormative structural injustice [42], which negatively impacts trans children [43,44]. These theoretical approaches underpinned the ethics and methodology of this study, which recognizes that gender diversity is neither pathological nor problematic and whose aim is not to prescribe but to describe and understand.

From this perspective, the knowledge and position of researchers are valuable resources that enrich the analysis of and reflective engagement with the data. Our research team is composed of two cisgender women and one cisgender man, all three of whom work as teaching and research staff in three public universities in Spain, have training in sexology and share the values of the trans-affirmative framework. One of the researchers has a trans daughter and is a member of the organization of families of trans minors that collaborated with this study.

In a broader research framework, whose aim is to analyze the experiences of the families of trans children and adolescents in three periods (before, during and after the transition) in order to show the consequences of the affirmative model, the present study focuses on the roles played by the different healthcare professionals who these families have been encountering in their transition process. This study focuses specifically on the experiences of those children and adolescents who at some point in their lives clearly stated “I am a boy” or “I am a girl” in opposition to the gender they were assigned at birth, and who have made a complete social transition, supported by their families, to live according to their gender identity, distinguishing them from those who show non-normative gender behavior, consider themselves non-binary or manifest other facts of gender diversity. In addition to social transition, in some cases, the children and adolescents in this study have achieved a legal change of name and/or sex (“legal transition”), and some of the pubescent children and adolescents have initiated endocrinological treatments (“medical transition”).

## 2. Materials and Methods

This qualitative study included 22 stories written by parents of trans children and adolescents aged from 4 to 18 years. All participants were members of the Naizen Association in the region of Basque Country (Spain). This association aims to support the families of trans children and youths, as well as to make their situation visible and defend their rights. All of the participants’ children have socially transitioned and, at the time of the study, lived in accordance with their expressed gender identity. The ages at which the children and adolescents transitioned ranged from 3 to 16 years; 14 were under the age of 10 years, and 8 were aged 10 years old or older at the time they made the transition. There were 13 were girls (assigned as boys at birth) and 9 boys (assigned as girls at birth). All of the parent participants were heterosexual and, except in one case, all of them lived as a couple at the time of transition. Of the 22 stories, 17 were written by the mother, 3 by the father and 2 by both parents. In most cases (18 of 25), the time between the transition and the writing of the detailed account requested in the present study did not exceed 2 years.

The sample was recruited by convenience sampling. All of the families in the association were invited to participate in the study by means of an advertisement. Anonymity and confidentiality were guaranteed. Participants did not receive any compensation, and the study was approved by the ethics committee of the University of the Basque Country.

Table 1 shows whether the story was written by the mother, father or both parents, and provides data on the age and educational level of the parents and the main characteristics of the children: the pseudonym used in the study, gender identity (indicated by the child/adolescent), age at the time of social transition, age at the time the story was written and cases in which medical transition has been initiated by puberty blockers and/or cross hormones.

Family members of the association were asked to voluntarily participate in the study by writing a story according to the following instructions: “Express through a detailed story the life processes of your trans child, and of the whole family, from birth to the present, before, during, and after transition. Recount what was experienced at home, in school, in the neighborhood, and in health services, along with your experience with administrations, associations, other families, etc. We are especially interested in anecdotes as well as literal expressions and even dialogues. Please refer to the best and easier aspects, as well as the worst and more difficult issues. The goal is to recount the issues that you consider significant. The text should cover about 8 pages (Times New Roman format, size 12, single spacing)”.

They emailed their stories to one of the researchers, who pseudonymized them before sharing them with the rest of the study team. The stories were written in Spanish. Both the analysis of the results and the subsequent discussion were carried out in Spanish, and the translation into English was performed later so that the article could be published in an international journal.

For the thematic analysis, we first collected the detailed stories. In terms of length, they ranged from 1665 to 5551 words.

The MAXQDA 22 program was used for the data analysis. MAXQDA is a professional software package that can be used for qualitative and mixed methods analyses, as well as for processing quantitative data after the categorization of discursive elements. Its use in qualitative studies, such as the present one, facilitates codification of the phenomenon of interest through the use of codes and categories, facilitating detailed analysis of the stories on both an individual and collective basis.

Two members of the research team were in charge of organizing the themes into categories and codes. After independent and simultaneous coding by two team members, the inter-coder agreement at the code level for the activated codes was found to be 66%. It should be noted that this percentage refers exclusively to inter-coder agreement regarding the presence of these codes in each of the documents analyzed. We believe that the observed disagreements were mainly due to the fact that the first evaluator grouped the content into larger segments. Thus, while the first evaluator coded a total of 71 segments of information, the second coded 90 segments.

Finally, the information was organized in a codebook based on three time points (categories): (1) Before transition: information about what the family experienced from birth to the first occasion on which the child’s/adolescent’s gender identity was explicitly disclosed; (2) During transition: information about the experience between the first explicit disclosure of the child’s/adolescent’s gender identity and the social recognition thereof; (3) After transition: information about what the family experienced after social recognition of the child’s/adolescent’s expressed gender identity up to the present.

We obtained a large number of codes, subcodes and emerging themes based on the different people and groups involved in the child’s life and the assessments made by the participants of the attitudes and responses of others (educational center, healthcare professionals, associations, peer group, administration, etc.).

In this study we focus on healthcare professionals and analyze information regarding the participants’ assessments of the responses and support received from these professionals in the three categories delineated above: before, during and after the transition.

At each of the three time points, we found different health professionals interacting with the child. Before transition, psychologists and sexologists played an essential role as a source of information and advice; during transition, other health professionals, such as pediatricians, became more important; and finally, once the transition had taken place and puberty was approaching, it was endocrinologists, psychiatrists and other professionals from gender units who played a more prominent role.

Table 2 shows the frequency with which the stories refer to the role of the different professionals in the three time points analyzed.

When analyzing the data, two large thematic blocks were found that were transversal to the three time points indicated (before, during and after transition) and were associated with the different agents (pediatricians, psychologists, psychiatrists, endocrinologists and sexologists). The first, which has been called positive support, was made up of the facilities, acceptance and support experienced by the children and adolescents, and their families, while the second, called difficulties, was made up of negative attitudes, lack of understanding and confusion with other issues, such as non-normative gender behaviors.

## 3. Results

We illustrate the results with excerpts that exemplify the different topics of analysis. After each fragment, in parentheses, we indicate the following (in order): pseudonym of the protagonist of the story, gender identity (recognized after transition), age at the time of the transition, age at the time the story was written (data regarding the latter does not appear in one of the cases, in which the adolescent died from suicide) and authorship of the story: father (F), mother (M) or both (F/M). As an example:


*“We went back to the psychiatric care center for children, and there began a path of useless treatments that not only did not solve the situation, but often worsened it with side effects: sleep disturbances, more sudden weight loss, nervousness…”*
(Irene, girl, 10, 12, F)

This fragment corresponds to a story that narrates the transition process of Irene (pseudonym), a girl (assigned as a boy at birth) who was 10 years old at the time she made the transition and 12 years old when the story was written; the story was written by her father (F).

### 3.1. Before Transition

Only 13 of the 22 stories analyzed mention health professionals before the transition, and they did so to point out that, in their search for information about what was happening to their child, the parents turned to these professionals, but they did not always receive the information they expected to obtain:


*“We consulted pediatricians, psychologists, sexologists… and the truth is that not even the professionals who work with children know about the subject; in fact, some of them only managed to confuse us even more.”*
(Ane, girl, 4, 5, M/F)

Parents who turned to pediatricians for a first opinion and information about how to act found that pediatricians had little knowledge on the subject:


*“The pediatrician had heard about the issue, but had no idea how to proceed so she agreed to look for information and call me to discuss it.”*
(Aitor, boy, 4, 4, M)

In six cases, families turned to psychology professionals for advice at an early stage. Only in one case was it noted that the professional was clear about the situation from the first moment:


*“When we returned from that holidays we decided to go to a child psychologist, without Eric, to explain to her what was worrying us. She didn’t know us at all; we told her everything and her words were: you have to mourn, your boy is a girl.”*
(Eric, boy, 4, 6, M/F)

In the other five cases, the professionals, sometimes with very little tact, considered the issue to be merely a matter of age such that the child or adolescent will “get over it”:


*“We went to a psychologist who did some tests and the result she gave us was that everything was normal, that there were girls who did not want to recognize and accept the issue of the period because they did not want to grow up. And that was the end of it.”*
(Unax, boy, 16, 18, M)

Moreover, some of the professionals gave specific guidelines to “avoid confusing” the child:


*“I went to a child psychiatrist and it was a really bad experience (…) He told me that we could not allow the child to do whatever he wanted, that letting him to do or play however he wanted at home was out of place. He told me to cut his hair because otherwise it might confuse him. Oh my God! My husband had long hair, would my son be seeing two mothers because of that?”*
(Ikerne, girl, 5, 10, M)

In one of the cases, it was stated that the parents went to the gender unit when their child was still far from puberty, looking for some guidance. In response, they were invited not to take any steps to recognize the expressed identity; instead, they were advised to make as few concessions as possible, mainly with the justification that it may be a “transitory phase” and a later “desistence” may occur:


*“The first thing he told us (endocrinologist) was that this topic is relatively new and very little is known. The rest I vaguely remember because it all seemed so surreal to me…. ‘It’s very small, it may be a phase. Although it is true that trans adults say they know it from a young age. Don’t force her one way or the other. Don’t deny her either. If you take the ribbon off one shirt, don’t take it off the next. Christmas gifts? Half for Aitor and half for Ane’.”*
(Aitor, boy, 4, 4, M)

Finally, in this pre-transition stage, six parents talked about sexologists as a source of information and advice. It should be noted that sexologists are professionals from other disciplines such as psychology, education or medicine who have specific postgraduate training in sexology and provide information, guidance, counseling and interventions in the face of difficulties and problems related to human sexuality. In all these cases, the referral to the sexologist was made by the family association, except in one case where it was the pediatrician who referred them. In all cases, the sexologist offered support that the families valued very highly:


*“We met a sexologist, pedagogue and psychologist, and a great person. Ugh! For me he was also someone indescribable, he was very helpful and supportive, he had to see a mother desperate to help her child and a father who was still anguished and resisted taking off the blindfold. With the small guidelines that he gave us, we took big steps”*
(Ikerne, girl, 5, 10, M)

### 3.2. During Transition

As soon as families realize that something is happening and that the best way to help their children is to support the transition process, they place themselves in the hands of professionals. At this point, the role of healthcare professionals was alluded to in 11 of the 22 stories, and pediatricians were the health professionals to whom 8 of the 11 cases turned to first.

At this point, families described the role of pediatricians in a friendlier way, as even those who did not know how to approach the issue showed a certain degree of humility and referred families to other professionals and associations:


*“The woman (pediatrician) had no idea what to do but her willingness to help and do what she could was wonderful. She made several calls but could not refer us to the gender unit since they had no idea of the protocol or anything, but she looked at Aiur and said: “Don’t worry, I will send you to the Unit”. And so it was.”*
(Aiur, boy, 16, ---, M)

Regardless of the profession (pediatrician, sexologist, psychologist, psychiatrist and even acupuncturist), parents considered the attitudes of those professionals who, at the moment of acceptance and beginning of the transition, listened to them without judging them, in an especially positive light:


*“Irati came back from that visit very happy, with a lot of energy and a lot of encouragement. She was able to give someone else what she felt, and she felt listened to and valued.”*
(Irati, girl, 14, 16, M)

On the contrary, the parents expressed great displeasure at the non-understanding attitude of some professionals, especially in the fields of psychology and psychiatry: *“They treated Amaia like a mentally ill person, it was heartbreaking.”* (Amaia, girl, 4, 6, M).

In some cases, it was the parents who needed and appreciated the help of a psychologist or other professional for themselves:


*“For a while I worked with the psychologist. Personally, she helped me because I was living in a very hard situation, and sometimes I did not know how to process it or how to order my own feelings and sensations towards my son. But, on the other hand, I have to say that the situation and the relationship with my son did not improve at all.”*
(Unax, boy, 16, 18, M)

### 3.3. After Transition

In 8 of the 22 stories, health professionals were mentioned after the social transition, in all cases making some reference to the gender units; these units played a very relevant role in the life of these families, especially for those children who are approaching or going through puberty.

Gender units were mentioned as an important resource that could allow body development to be harmonious with the adolescent’s identity: 


*“That’s why a few months ago we went to the gender unit, so that when she needed to Haize could start with blockers so she wouldn’t develop male characteristics that she didn’t want and that would make her life difficult.”*
(Haize, girl, 7, 12, M)

Adolescents and their families expect endocrinological treatment from gender units; however, the protocols require prior psychiatric and psychological treatment, which greatly lengthens the process and is often experienced by the adolescents and their families as an obstacle:


*“At that time, the appointments at the gender unit also begin. The mandatory appointments are with the psychiatrist and then the endocrinologist. Now, after 2 years of consultations, Irati is already receiving hormonal treatment and is in a very nice moment, starting to change and seeing how she transforms, step by step.”*
(Irati, girl, 14, 16, M)

One family reported having received adequate and cordial treatment in the gender unit, but the remaining families gave very negative evaluations of the treatment received there:


*“When it seemed that everything was getting on track on a personal, family level… we ran into the obsolete, to call it “politely” in some way, protocol of the gender unit.”*
(Martin, boy, 15, 17, M)

Those who came to the gender units at puberty reported that their experience had not been good and complained about the initial reception and the months spent waiting to obtain appointments (up to 6 months for the first appointment). They then spent up to a year trudging through psychology and psychiatry consultations before seeing an endocrinologist, which was, in fact, the specific care they needed:


*“For a year we have been back and forth to the unit (100 km) several times; first with the psychologist, when he gave the go-ahead with the psychiatrist, and when the latter gave the go-ahead with the endocrinologist… In other words, a year to get to the endocrinologist, which is what my adolescent son was demanding as his priority since he told us about it.”*
(Martin, boy, 15, 17, M)

In three of the eight stories, the parents described the first appointments in the gender unit with psychologists and psychiatrists as like being in court or in front of a tribunal:


*“There were three doctors sitting there: a psychologist, I think I remember, a psychologist and a psychiatrist. It looked like a trial; the psychologist started asking Aiur a lot of questions, she only had to ask him what size underpants he wore. The psychologist kept writing things down and the psychiatrist looked us up and down without saying anything. It was very uncomfortable, to be honest.”*
(Aiur, boy, 16, ---, M)

In this process, with the exception of one case, the rest reported a poor relationship with both psychiatrists and endocrinologists:


*“The psychiatrist (…) as it had happened other times, referred to him as feminine, even though there was already a medical record of my son where his chosen name already appeared. He underwent tests and we thought that the issue was starting to move forward. But no, again we encountered more obstacles.”*
(Unax, boy, 16, 18, M)

In many cases, health professionals postpone the hormonal treatment despite the urgency and discomfort expressed by the adolescent, which leads the families to turn to other services, private or public, in other autonomous communities, which do respond to their demands:


*“Well, my son could not wait any longer, and knowing that there was a possibility that they could take the treatment to another gender unit in another autonomous community, we could not wait any longer.”*
(Martin, boy, 15, 17, M)

The story of one of the mothers is particularly harsh. Without blaming anyone for the death of her son, who committed suicide, she wonders what would have happened if they had started the hormone treatment they were demanding earlier that the gender unit’s psychologist team insisted on postponing:


*“Aiur had been waiting for his hormonal treatment for a year, after several meaningless consultations, with stupid questions and lies and telling Aiur when he told the psychologist that he had a very bad time when he had his period and with his breasts not to focus on that and to look for alternatives… Someday, my husband and I will return there; we will look for the psychologist, and we will only tell him: Do you remember us and our son? Do you remember that you told him not to focus on his period or breasts and to look for alternatives? Well, look… he listened to you… do you like the alternative he took?”*
(Aiur, boy, 16, ---, M)

Regarding the gender units, some stories stated that, in recent years, the service they provide has been improving, and many professionals are changing their approach to trans children and youths:


*“We have started to go to the new trans care unit, and his predisposition is so different… Before my son always went to the clinic full of resentment and frustration; now he is happy, cheerful, brave and full of optimism.”*
(Daniel, 14, 15, M)

Moreover, one of the stories explains the fundamental role that pressure from family associations has played in bringing about a change in the functioning of these gender units:


*“As a result of all this, the Director General of Healthcare became personally involved in the issue. They had several meetings with the Association, the Director of Equality and other professionals, where they became aware that the Unit as it functioned had to change. I still remember the words she said to my son: “Know that thanks to you and how badly we have done things with you, we have seen that the unit is going to change.”*
(Unax, boy, 16, 18, M)

Overall, all of the stories mentioned the lack of knowledge and training of health professionals, considering it one of the main barriers the children have encountered:


*“Now I look back and see everything that at such a young age has happened to her, and I think that if there had been more information and more knowledge of the subject, my daughter would not have suffered so much. Because seeing the bitter crying of a 5-year-old girl who tells you that sometimes she just feels like crying hurts. Consulting the pediatrician and being told not to worry, and that the only thing he knows (and he has no desire to learn) is that with time it will pass as if it were a cold, is very sad. And when you go with her to the office, he does not even know how to act! By God, that my daughter gets the flu just like the rest of the children! Don’t look at her strangely! And the pediatrician is astonished to know that my daughter doesn’t go to the psychiatrist… to the psychiatrist, why? My daughter is not sick! Look, if you want, I will inform you! But he doesn’t even listen. These situations generate a lot of impotence; why don’t they bother to learn?”*
(Euri, girl, 6, 7, M)

## 4. Discussion

The aim of this study was to explore the role that healthcare professionals play in supporting trans children, trans adolescents and their families. Previous research has indicated that healthcare professionals are key to guide families in their transition processes. Therefore, they are expected to provide support. However, previous research does not seem to provide evidence of their positive support, instead documenting their lack of knowledge and the need for more inclusive healthcare environments specific to trans youths and families [41]. Our study shows that: (1) all families searched for healthcare assistance at some point; (2) families searched for different types of healthcare professionals based on the social transition stage (before, during or after), moving from psychologists and sexologists to medical professionals (pediatricians or gender units); (3) in many cases, the professionals lacked specific knowledge; and (4) depending on the professional, their attitude facilitated or hindered the transition and psychological well-being of these children and adolescents, and their families. Table 3 shows the issues identified in the results.

In our study, the time when families mentioned health professionals the most was before transition (59.1%), at that stage they do not understand what is happening. During the social transition, in half of the stories (50%), the families turned to professionals to help them in the face of the confusion generated by the already-explicit situation. After the social transition, only a little more than one-third (36.33%) of the stories made reference to health professionals; those stories were almost exclusively of adolescents, above all, referring to gender unity and the need to access endocrinological treatments.

It is of interest to note that the families, before and during social transition, tended to search for professionals who may be experts in gender identity issues and may be able to advise them and support their children, similar to previous studies [30]. Interestingly, however, once the expressed identity is accepted by the family and the social transition has already been made, very few parents mentioned health professionals in the case of prepubescent children; this is not because the children no longer visit pediatricians or other physicians for other health issues but instead probably because parents do not experience particular difficulties regarding their prepubescent children’s gender identity after social transition, and they no longer need the help of professionals for this issue or do not consider it relevant to talk about. However, in the case of those who are entering or are already in puberty, families request medical help at the gender units to avoid unwanted pubertal changes and facilitate physical changes according to their affirmed gender identity.

Regarding families’ experiences with healthcare professionals, in many cases, the families tended to report a great sense of lack of knowledge, as previously indicated [31,45]. In this sense, many of the professionals consulted by these families considered the children’s behavior to be only a phase that would probably change across the years, and they encouraged parents not to make major gender concessions, discouraging affirmative support. This professional positioning, which invites watchful waiting, was typical of earlier times [46]. However, in light of recent studies showing the advantages of the affirmative model from early childhood [12,13,16], the watchful waiting approach could lead to negative consequences.

When families obtain awareness about what is happening with their children and decide to support them in their transition, the role of pediatricians becomes especially relevant. Our participants described how they had received friendly treatment from pediatricians, although their role was often reduced to facilitating families’ access to other professional resources, associations, etc., in line with what has been reported in other studies [35].

Gender units and their professionals were mentioned, especially in the time after transition, particularly when puberty had already begun and endocrinological treatments were required. With only one exception, the experience of the families in the gender units can be characterized as negative in light of what was expressed in their stories. Waiting lists and excessive bureaucracy were the first difficulties they encountered. In addition, parents felt that psychiatrists and endocrinologists tried to postpone endocrinological treatment, acting with a slowness and prudence that did not correspond to the urgency expressed by the affected adolescents. Parents lived this experience with discomfort, resulting in confrontations between families and professionals similar to those described in previous studies [25]. Various experts recommend caution in the overall management of cases and may even favor delaying treatments that have irreversible consequences [47]. However, in light of the stories analyzed herein, delaying hormone treatment, far from being an act of prudence, could be considered an iatrogenic decision with devastating effects on the psycho-social development of these adolescents. New models of care focusing on transaffirmative care have led to greater accessibility to care for this group [48] and have improved their well-being, as reflected in one of the stories in this study in which the transition began more recently; this is in line with recent evidence demonstrating the benefits of both pubertal blocking and gender-affirming treatment on the health care of trans people [4,49].

In contrast to the negative practices carried out by some health professionals, sexologists seem to play a more positive role in supporting the families of trans children before, during and after the transition. As expressed by the participants in our study, sexologists listen and are empathic to their fears, do not judge them and explain what is happening rigorously, without judging their children’s behavior. Sexologists were not the only ones to show these attitudes; however, they were also found in some pediatricians, psychologists and even in an acupuncturist. Nevertheless, of all the professionals to whom these families turned, sexologists were the only ones whose performance was positively evaluated in all cases. This reinforces the importance of proper training and knowledge of human sexuality to provide effective support [36].

Proper training of sexologists is obvious, but in most cases, other professionals, such as pediatricians or psychologists, do not receive it. We have not found any other research that talks about the role of sexologists in supporting trans children and adolescents, and in view of the very positive assessments made by the families in our study, we believe that it may be instructive to delve deeper into the possibilities offered by sexological conceptualization and intervention in this field.

## 5. Conclusions

This study had several limitations. First, the participating families are all members of the same association of families of trans children and adolescents, which probably leads them to having a shared vision. It would be instructive to extend the study to families from other autonomous communities, as well as to families from other associations and to those that do not belong to any association, in order to explore other experiences and perspectives. In addition, the stories of the lived experiences are influenced by the fact that more mothers than fathers participated in the study (19 mothers and 5 fathers). Since the study is based on retrospective information, there may be some recall bias, given the high emotional charge of the experiences being recounted, so it would be convenient to carry out other studies using other instruments that allow the recording of specific behaviors and/or situations on a day-to-day basis.

Nonetheless, this is one of the first qualitative studies carried out in Spain to document the experiences of the families of trans children and adolescents in relation to the healthcare setting. Moreover, to our knowledge, it is the first in which the target population was differentiated from other related groups, such as those with non-normative gender behaviors or the non-binary population. All of these groups share some characteristics, especially the experience of stigma and exclusion, and therefore they all merit attention. However, as these are different issues, it is essential to know the specific experiences and needs of each group in order to be able to adapt responses and actions to better support them.

Future studies should delve more deeply into the role played by healthcare professionals during and after the transition process in order to provide indicators that will allow us to improve the care and counseling that they provide. However, it will be necessary to address this issue through longitudinal designs because such studies allow follow-up from childhood through adolescence to adulthood, thus providing a broader and more dynamic view of the transition processes and the role that should be played by the different professionals involved. There is a need for studies focused on the capacity and preparation of healthcare professionals to address aspects of gender identity and diversity, given that this is a key issue for the provision of adequate support in collaboration with families. Training these professionals in sexuality and diversity is probably the only way to ensure care for trans children and adolescents, and their families, as it does not depend solely on issues such as the professional’s attitudes, current policies or professional “fashions”. Finally, finding the right balance between the speed demanded by some adolescents and their families and quiet attention, which allows for a deeper understanding of the specific needs of each person and thus the establishment of appropriate and individualized treatment, is one of the great challenges faced by professionals [48].

Qualitative studies focused on the experiences of trans children and adolescents, and their families, are a fundamental tool for achieving this balance, which will only be possible by listening to their needs.

## Figures and Tables

**Table 1 healthcare-12-00974-t001:** Participants’ characteristics.

Author	Age	Education Level	Children’s Pseudonym	Gender Identity	Transition’s Age	Age	Puberty Blockers	Cross-Sex Hormones
Father	38	Technical	June	Girl	3	4	-	-
Mother	43	Technical	Aitor	Boy	4	4	-	-
Both parents	31/36	Technical	Mikel	Boy	4	4	-	-
Mother	45	Secondary	Jon	Girl	4	5	-	-
Mother	37	Postgraduate	Ane	Girl	4	5	-	-
Mother	37	Technical	Amaia	Girl	4	6	-	-
Both parents	39/38	Undergraduate	Eric	Girl	4	6	-	-
Mother	35	Undergraduate	María	Girl	4	7	-	-
Mother	45	Secondary	Paul	Boy	4	8	-	-
Mother	45	Postgraduate	Amaiur	Boy	5	8	-	-
Mother	42	Technical	Ikerne	Girl	5	10	-	-
Mother	37	Technical	Euri	Girl	6	7	-	-
Mother	38	Undergraduate	Mael	Boy	7	8	-	-
Mother	45	Secondary	Haize	Girl	7	12	-	-
Father	48	Undergraduate	Irene	Girl	10	12	-	-
Mother	51	Primary	Julene	Girl	11	17	Yes	Yes
Mother	48	Undergraduate	Daniel	Boy	14	15	-	-
Father	52	Postgraduate	Erin	Girl	14	15	Yes	-
Mother	49	Undergraduate	Irati	Girl	14	16	Yes	Yes
Mother	50	Undergraduate	Martin	Boy	15	17	-	Yes
Mother	58	Technical	Unax	Boy	16	18	-	Yes
Mother	43	Primary	Aiur	Boy	16	---	-	-

**Table 2 healthcare-12-00974-t002:** Stories mentioning the role of different professionals before, during and after transition.

		Frequency		
		Before	During	After
	Sexologist	6	4	2
	Psychologist	6	2	1
	Psychiatrist	1	0	0
	Pediatrician	3	8	0
	Others (acupuncturist)	0	1	0
**Gender Unit**	GU generic reference	0	2	5
	GU Endocrine	1	1	6
	GU Psychiatrist	0	0	4
	GU Psychologist	5	1	3
**Documents with codes**		13 (59.1%)	11 (50%)	8 (36.33%)

**Table 3 healthcare-12-00974-t003:** Most relevant issues identified.

**Before**	**Most professionals:** ⎛ **Lack of knowledge** ⎛ **Guidelines for non-recognition of expressed identity** ⎛ **Arguments: a matter of age, transitory phase, desistences…** **Sexologists:** ⎛ **Source of information and support**
**During**	Positive evaluation ⎛ When professionals listen to them without judging Negative evaluation ⎛ When professionals show an attitude indicating a lack of understanding ⎛ Lack of understanding is especially pertinent in the fields of psychology and psychiatry Pediatricians are particularly relevant ⎛ Refer families to other professionals and associations Some parents need help themselves
**After**	Gender units ⎛ Highly relevant role, especially in puberty ⎛ Resources that could allow the necessary body development ⎛ Mostly negative evaluations: ♠ Poor relationship with professionals ♠ Lack of knowledge and training ♠ Protracted time spent waiting ♠ Required hormonal treatment postponed despite the urgency, as experienced by adolescents ♠ Imposed psychiatric and psychological treatment that was not requested ⎛ In recent years, there have been improvements in the services provided

## Data Availability

Data will be made available on request.

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
