# Peer review of "The Role of Healthcare Professionals in the Transition Process: A Qualitative Study of Families of Trans Children and Adolescents"

_healthcare, 2024, doi:10.3390/healthcare12100974_

Round 1

Reviewer 1 Report

Comments and Suggestions for Authors

This paper tackles an important topic about how families navigate healthcare systems in relation to transgender youth. The qualitative study helps understand the experiences of these families in a different way than simply quantitative studies. 

It would be helpful to have more information about the healthcare structure in Spain. How are referrals made? Are there access barriers? The structure and protocols of the gender units are not well described. Do they follow WPATH standards of care? This information would be helpful in understanding the generalizability of this study. Authors mention changes were made to gender units related to this feedback - is this study still relevant given the evolution of gender units described? Are there cultural factors at play that would be helpful to describe?

Author Response

Thank you very much for your review, please read the attached file with our answers and corrections. 

Reviewer 2 Report

Comments and Suggestions for Authors

This study aims to explore the role of healthcare professionals in supporting transgender youth during the transition process. The authors describe a qualitative study comprised of retrospective accounts written by parents of transgender children and adolescents describing various aspects of their transition. The authors focus their analysis specifically on parents' reports of interactions with healthcare providers.

Major comments: 

1. The introduction needs to be significantly edited. 

- Authors need to consistently use the term "gender identity" rather than gender to refer to youth's identity.

- "Trans" is colloquial - the more professional term is transgender and should be used in a scientific manuscript.

- Rather than referring to genitalia it would be more appropriate to state that the sex assigned at birth is based on biology and anatomy.

- Literature does not support the claim that transgender identity universally emerges around the time of puberty. There are multiple peer-reviewed studies documenting various ages of gender identity recognition, including in adulthood and should be included.

- The authors state there is "no consensus on how to care for trans children," whereas there are two sets of guidelines that provide such a consensus. The Endocrine Society and WPATH both have published guidelines addressing gender identity in youth. The authors need to include these in their introduction.

- Rather than using terms such as gay or lesbian it would behoove the authors to refer to sexual attraction using more accepted terminology.

- LGTBIQ is not the preferred acronym; would recommend using instead LGBTQIA.

2. Materials and methods

- Clarify which participants "lived as a couple," the parents or the children.

- The timing of before, during, and after transition is not described clearly. Recommend clarifying these time points.

3. Results

- Currently the community prefers to use the term "died from suicide," rather than "committed suicide." Suggest adjusting terminology.

- Sexologist is not a universally-familiar type of provider. Recommend the authors define this term and provide background information into what type of provider this is.

4. Discussion

- Recommend creating a visual representation of the thematic elements identified, either in table, chart, or graphic to more clearly show the results.

- The first recommendation for future studies does not reflect the findings of the current study. This manuscript does not describe the transgender youth's experiences at all, however the authors posit that their study supports future research into quality of life and mental health of transgender youth. Although this is needed, their study does not support this.

- The authors state that there must be a balance between desire for medical treatment for transition and speed. Recommend they become familiar with the guidelines published by the Endocrine Society and WPATH in this regard.

Minor comments:

- Why were the narratives only requested from parents, rather than the youth themselves? It would expand the study to have comments from the individuals with gender dysphoria.

Comments on the Quality of English Language

- Sentence structure could be improved; many sentences are quite long and would be better served by splitting them up.

- Line 155-156 has wording errors.

- Throughout recommend using transgender rather than trans.

- Not necessary to state the reason for translation later into English. If hoping to publish in an international journal need to ensure translation and grammar is appropriate.

Author Response

Thank you very much for your review, please read the attachment with our answers and corrections. 

Reviewer 3 Report

Comments and Suggestions for Authors The bibliography predominantly cites works by authors who actively support the transition even in extrascientific fields (e.g. F. Asley) and constantly cites works that contest papers that recommend caution in proceeding quickly with the transition. Reference 9 (... Grupo GIDSEEN..) is contested by reference 11, but there is no trace of the response to this letter from the authors of reference 9, which is published in the same issue of the journal immediately after ref 11. Similarly, papers like the one of Schumm and Crawford (https://doi.org/10.1177/002436391988479) which criticizes the overly optimistic data of the papers on the transition are not discussed. It could be my subjective impression, but the language and the overall structure and in particular of the introduction and discussion of the paper seem aimed more at the a priori affirmation of a cultural position than at research and discussion.

Author Response

Thank you very much for your review, please read the attachement with our answers and corrections. 

Reviewer 4 Report

Comments and Suggestions for Authors

1.      Text detailing the authors’ positionalities should be added. Specifically, as expected with qualitative research, the authors should particularize: a) why they elected to study this topic?; b) their relationships with this topic; c) their expectations about study outcomes; and d) how these expectations were challenged/affirmed by the key findings.

2.      Additional information should be given about the data analysis program (MAXQDA 22).

3.      Line 34: remove “the” before “adolescence”

4.      Line 178: use “administrations” (lowercase)

5.      Line 435: delete “those” before “of adolescents”

6.      Line 486: use “empathic” rather than “empathetic”

7.   Lines 510-512: The sentence beginning "Nonetheless..." is awkward and should be modified for clarity. 

Comments on the Quality of English Language

A few minor edits have been suggested. 

Author Response

(The authors gave the same response as above.)

Round 2

Reviewer 2 Report

Comments and Suggestions for Authors

1. While I appreciate the edits made by the authors in response to suggestions from reviewers, I still question the novel nature of their study, and the applicability to the worldwide healthcare community.

2. There remain grammatical errors, specifically in the sections added since the original manuscript was submitted.

3. Although this study was conducted in Spain, if the authors wish for their data to be generalized to the worldwide population it would be beneficial to more clearly describe the current practice for care of this population in Spain, and contrast how this may differ in other parts of the world. 

4. I still suggest using the full term, "transgender," or if the authors desire "gender diverse," rather than "trans."

Comments on the Quality of English Language

See above.

Author Response

Dear Reviewer. Thank you again for your input and suggestions, all of which are very valuable and are helping us to reflect on the content of our study and improve it substantially.

  1. While I appreciate the edits made by the authors in response to suggestions from reviewers, I still question the novel nature of their study, and the applicability to the worldwide healthcare community.

The authors of this research understand that this is a novel study since there are very few academic publications that collect the experiences with health professionals of families with trans children who have transitioned. We consider it very valuable to collect the voices of these families through academic research to advance knowledge about their experiences and needs; knowledge that makes it possible to think and design health care that listens to their needs, their demands, their discomforts, etc. in order to take better care of them.

  1. There remain grammatical errors, specifically in the sections added since the original manuscript was submitted.

The article and modifications have been proofread again by a native translator. We hope that there are no more grammatical errors.

  1. Although this study was conducted in Spain, if the authors wish for their data to be generalized to the worldwide population it would be beneficial to more clearly describe the current practice for care of this population in Spain, and contrast how this may differ in other parts of the world.

In response to another reviewer's suggestion, we have already included a paragraph pointing out that there are differences between countries (line 77 to 84) and refer to some papers that cover them. What you suggest, although particularly interesting and potentially relevant, is beyond the scope of this paper, since going into the differences between countries in greater depth would entail writing at least another full article.

  1. I still suggest using the full term, "transgender," or if the authors desire "gender diverse," rather than "trans."

For us, it is important to keep the term trans. Not only because, as we explained in the previous review, trans and transgender are currently used interchangeably in the field of social sciences, but also because, according to other authors, doing so avoids misunderstandings derived from the different implications that transgender has in one or the other language .

Thus, according to van Anders et al. (2019), although the term trans is sometimes considered more informal/less formal, and it is a term that people use to identify themselves or their experiences, it is also used as a term for an academic discipline (trans studies) and as an academic subject.  These same authors define transgender as a synonym of trans.

In Spanish, the term transgender does not have exactly the same meaning as in English. Rather, it functions as "a term that denotes a willingness to move away from the dominant paradigm of understanding gender, considered pathologizing, binarist and reifying the male and female categories" (Coll-Planas and Missé, 2015, p. 40). That is, it implies a political positioning and a distancing from cisheteronormativity that not all trans people have, even less so in the case of the children who are the object of our study.

Although the article is published in English and in an international journal, we understand that trans makes it easier for Spanish-speaking people to understand the issue we are referring to.

References:

Coll-Planas, G. & Missé, M. (2015). La identidad a disputa. Conflictos alrededor de la construcción de la transexualidad. Papers 100 (1), 35-52.

van Anders, S. M., Galupo, M. P., Irwin, J., Twist, M. L. C., Reynolds, C. J., Easterbrook, R. B. C., & Hoskin R. A. (2019). Talking about transgender experiences, identities, and existences. Link: https://docs.google.com/document/d/1iHodSA16oP0itTjZPkB5tslBjMHOiMdy9lt9zmTPKPs/edit?usp=sharing
